# Control Strategies Applied to a Heat Transfer Loop of a Linear Fresnel Collector

Alaric Christian Montenon [1,*,†] and Rowida Meligy [2,†]

1 Energy, Environment and Water Research Center, The Cyprus Institute, 20 Konstantinou Kavafi Street, Aglantzia 2121, Cyprus
2 Mechatronics Department, Faculty of Engineering, Helwan University, Cairo 11795, Egypt; rowida.meligy@h-eng.helwan.edu.eg
* Correspondence: a.montenon@cyi.ac.cy; Tel.: +357-222-08-672
† These authors contributed equally to this work.

**Abstract:** The modelling of Linear Fresnel Collectors (LFCs) is crucial in order to predict accurate performance for annual yields and to define proper commands to design the suitable controller. The ISO 9806 modelling, applied to thermal collectors, presents some gaps especially with concentration collectors including LFCs notably due to the factorisation of the incidence angle modifiers and the fact that they are considered symmetric around the south meridian. The present work details the use of two alternative modellings methodologies based on recorded experimental data on the solar system installed at the Cyprus Institute, in the outskirts of Nicosia, Cyprus. The first modelling is the RealTrackEff, which is an improved ISO9806 modelling, and the second is constructed using the CARNOT blockset in MATLAB/Simulink. Both models include all the elements of the heat transfer fluid loop, i.e., mineral oil, with a tank and a heat-exchanger. First, the open loop's studies demonstrated that the root mean square on temperature is 1 °C with the RealTrackEff; 2.9 °C with the CARNOT and 6.3 °C with the ISO9806 in comparison to the experimental data. Then, a PID control is applied on the experimental values in order to estimate the impact on the outlet temperature on the absorber and on power generation. Results showed that the error on the estimation of the heat absorbed reaches 32%.

**Keywords:** linear fresnel reflector; modelling; control; heat transfer fluid

## 1. Introduction

Linear Fresnel Collectors (LFCs) are one of the four main concentration technologies available [1], which are divided into two categories: the point focusing (central receivers [2] and dish-Stirling [3]) and the linear focusing (LFCs [4] and parabolic trough collectors [5]). The latter ones rely on linear receivers tubes [6–8] assembled in series. They can be evacuated or atmospheric, sometimes mixed [9]. The tubes are absorbers that receive the concentrated solar income reflected by the primary optics or reflector. It transfers the heat to a fluid in motion, a heat transfer fluid (HTF), sometimes for direct steam generation by evaporating water [10]. The reflector is composed of numerous (almost) flat mirrors [11] moving on a single axis to track the sun to reflect the DNI (Direct Normal Irradiance) on the linear receiver. Usually, the LFCs are topped by a secondary mirror above the tubes to harmonise the distribution of the solar flux around the absorber wall [12]. While point-focusing technologies are suitable for electricity generation at high temperatures (>600 °C), the LFCs are perfect candidates for mid-temperature levels (150–450 °C), especially for heat-processing purpose [13]. Although they require more land usage, the LFCs compared to other concentration technologies reportedly offer lower investment costs [14]. However, the reliability of the technology shall be supported by the accuracy of its annual yield prediction for techno-economic considerations [15]. To this end, the ISO9806:2017 is often used to predict the quasi-dynamic behaviour of solar thermal collectors [16].

It is also commonly used for LFCs [17]. TRNSYS uses the EN 12975-2, which is similar [15,18]. However, these methods present some flaws when applied to real experimental values. One of the flaws concerns the factorisation of the incidence angle modifiers (IAMs). Multi-linear regression methods permit to better approximate the quasi-dynamic behaviour of receivers [19,20]. Other models consider the internal energy variation in the absorber [21,22]. Many of the aforementioned models consider both linear and quadratic heat losses; however, [23] also considers the bi-quadratic heat losses factor. Modelling is crucial in terms of controls, especially considering the HTF flow, due to safety reasons. HTFs operate nominally at a certain range of temperatures for pumping purposes. Indeed, the HTF must be viscous enough and not overheated to avoid accelerated ageing and high pressure in the vessels. When simulating the operation of an LFC on an annual basis, these parameters shall be carefully taken into account. When leading a techno-economic study on larger systems approach, these particular but crucial points are difficult to be take into account, in spite of the fact that they play an actual role on the lifespan of the solar facilities. In the present study, the LFC at the Cyprus Institute is considered [24], which is operating up to 180 °C with the purpose of cooling an adjacent building, the Novel Technologies laboratory (Figure 1, [25]). More than 50 days of experiments have been registered with a time step of 15 s to 30 s. This allowed to determine the ISO modelling that was best fitting the experimental data, including the mass flow, the inlet and outlet temperatures, the ambient temperature and the DNI. The Cyprus Institute developed its own modelling, namely the RealTrackEff, which takes into account the asymmetric behaviour of the collector in terms of IAMs. This model has been compared to a third one: the CARNOT model, which is based on the model proposed by [26]. The control problem of LFC is concerned with forcing the output temperature to follow the reference, despite the existence of disturbances. A PID (Proportional Integral Derivative) to regulate the outlet temperature of the HTF has been implemented on the RealTrackEff, which is the most accurate model. This PID has been then applied to the two other modellings in order to evaluate its response and deviations.The development of the work follows the flow diagram in Figure 2. Section 2 presents the three modellings and the PID controller. Section 3 details the results of the application of the PID controller on experimental data. The paper ends with the conclusions in Section 4.

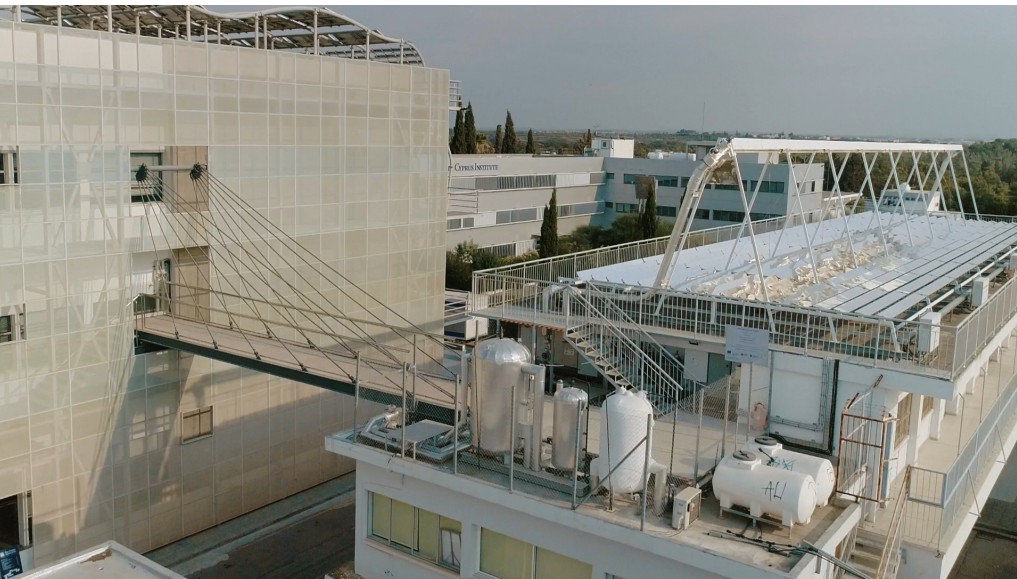

**Figure 1.** The Linear Fresnel Collector at the Cyprus Institute (**right**) and the Novel Technologies Laboratory [25] (**left**).

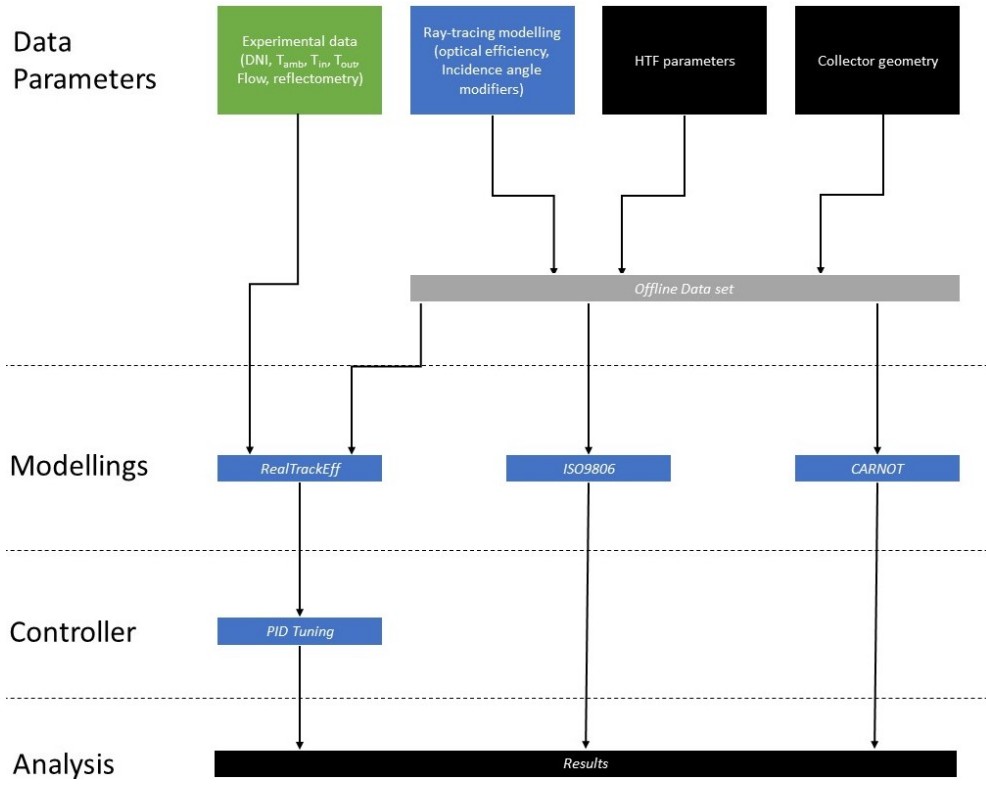

**Figure 2.** Flow diagram of the comparative modellings.

## 2. Materials and Methods

### 2.1. Modelling

The current section develops the modelling that has been used in order to predict the instantaneous power absorbed by the HTF at the Cyprus Institute at the level of the absorber: the ISO9806-based model [16], the RealTrackEff [27] tuning and the CARNOT modelling [28]. The north–south oriented receiver is made of 8 evacuated tubes in series, as detailed in Table 1, for a global length of 32 m. The outer diameter of the tube 70 mm. The vacuum is ensured with concentric borosilicate glass pipes of diameter 125 mm (Figure 3). The instantaneous power absorbed by the HTF, $\dot{q}$ (W), is:

$$\dot{q} = \rho \cdot \dot{V} \int_{T_{in}}^{T_{out}} Cp(T) \cdot dT \tag{1}$$

where:

- $\dot{V}$ (m$^3 \cdot$ s$^{-1}$) is the volumetric flow;
- $T_{in}$ (°C) is the inlet absorber temperature;
- $T_{out}$ (°C) is the outlet absober temperature.

**Table 1.** Main parameters of each of the eight evacuated tubes in series (model HCEOI12, data Archimede Solar Energy).

| Item | Value | Materials |
|---|---|---|
| Absorber tube diameter | 70 mm | |
| Absorber tube thickness | 2 mm | Stainless steel |
| Absorber tube length | 4.06 m | |
| Glass tube diameter | 125 mm | |
| Glass tube thickness | 3 mm | Borosilicate glass |
| Glass tube length | 3.9 m | |

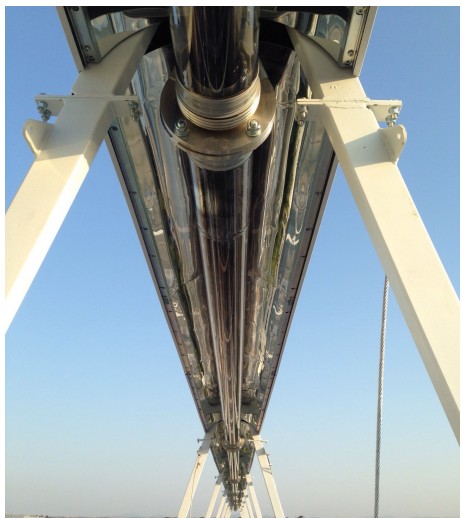

**Figure 3.** Receiver of the LFC, 32 m long with eight evacuated tubes in series.

Temperatures $T_{in}$ and $T_{out}$ and the volumetric flow $\dot{V}$ have been recorded with a sample time of 15 s or 30 s during 50 days of experiments, with an accuracy of $\pm 0.31\,°C$ (Thermocouple PT100 type) for the temperature and $\pm 0.75\%$ on the volumetric flow (Prowirl F200 model). The specific heat capacity $Cp$ ($J \cdot kg^{-1} \cdot K^{-1}$) and the mass density $\rho$ ($kg \cdot m^{-3}$) are given by the manufacturer's datasheet (Duratherm 450, mineral oil type), as a function of the temperature ($T$, $°C$):

$$\begin{cases} Cp(T) = 3.0360084 \cdot T + 2026.7 \\ \rho(T) = -0.68348 \cdot T + 878.20 \end{cases} \tag{2}$$

In the next parts, 3 different models are proposed to fit the estimated real power absorbed as in Equation (1).

2.1.1. ISO-Based Modelling

The first model to be used is the ISO9806 [16]. It has been defined for solar thermal collectors and is adapted to solar concentration technologies as follows:

$$\frac{\dot{q}}{A_{ref}} = \phi_r - a_1 \cdot (T_{amb} - T_{HTF}) - a_2 \cdot (T_{amb} - T_{HTF})^2 - a_5 \cdot \frac{dT_{HTF}}{dt} \tag{3}$$

where:

- $\phi_r$ ($W \cdot m^{-2}$) is the radiative income on the outer absorber wall, reflected by the primary and secondary reflectors;
- $A_{ref}$ is the area of the primary optics (184.32 m$^2$), made of 288 mirros (0.32 m × 2 m each [29]);
- $T_{amb}$ ($C$) is the ambient measured temperature;
- $T_{HTF}$ ($C$) is the average between $T_{in}$ and $T_{out}$;
- $a_1$ ($W \cdot m^{-2} \cdot K^{-1}$), $a_2$ ($W \cdot m^{-2} \cdot K^{-2}$), $a_5$ ($J \cdot m^{-2} \cdot K^{-1}$) are the heat loss coefficients that are relevant for concentration technologies with evacuated tubes in the ISO9806.

The model based on the ISO9806 does not consider diffuse radiation, as that has no benefit at all on the concentration technologies. Nor does is consider the wind effect, as in the present case, the absorber is enclosed in vacuum. In the first case for the classic ISO9806 model:

$$\phi_r = \eta_0 \cdot IAM(\theta_L, \theta_T) \cdot G_b \tag{4}$$

where:

- $\eta_0$ is the nominal optical efficiency;

- $IAM$ is the incidence angle modifier;
- $\theta_L$ is the longitudinal angle;
- $\theta_T$ is the transversal angle;
- $G_b$ is the measured direct normal irradiation (DNI, W $\cdot$ m$^{-2}$).

The $IAM$ and $\eta_0$ have been calculated in details with the use of Tonatiuh software with an accurate modelling of the collector, including all components [29].

### 2.1.2. RealTrackEff

The model based on ISO has been further developed as follows for RealTrackEff:

$$\phi_r = \eta_0 \cdot \eta_{cleanness} \cdot IAM(\theta_L, \theta_T) \cdot \Pi(\theta_L, \theta_T) \cdot G_b \tag{5}$$

where:

- $\eta_{cleanness}$ is the average cleanness state of the primary optics;
- $\Pi$ is a polynomial function of the incidence angles ($\theta_L, \theta_T$).

The cleanness state is defined as the ratio between: (i) the average reflectometry measured on 32 points on the primary optics with a reflectometer and (ii) the maximum cleanness state ever measured in practice. The polynomial $\Pi$ adding resulted in the observation of an asymmetric behaviour of the power throughout the day. Indeed, while it should be with a symmetric shape centred around the solar noon, it appeared that the power in the afternoon was not the same as for the morning [15]. Such effects are due to the real mirror shape (not perfectly parabolic nor symmetric), and the non-uniform tracking efficiency (friction, backlash, etc.) along the day cannot be rendered by the ISO9806 model.

### 2.1.3. CARNOT Modelling

The second LFC model is developed by means of a quasi-dynamic testing method which is implemented using the Conventional And Renewable eNergy Optimization (CARNOT) Blockset [26,30]. CARNOT is an open-source toolbox for MATLAB/Simulink developed by the Solar-Institute Jülich of FH Aachen, Jülich, Germany. It is a tool for the simulation of various components of heating systems such as solar collectors, thermal storage tanks, chillers, heat exchangers, pumps, pipes, etc. Codes are implemented in C language and linked to the simulation environment via an S-function. Regarding the mathematical model presented in this paper, the CARNOT model of a parabolic trough collector is modified to simulate the performance of an LFC [8]. Its model is a one-dimensional multi-node model which is divided into "N" nodes, where the flow is equally distributed among them, and the energy balance for every node is considered as:

$$C_{col} \cdot \frac{dT_{HTF}}{dt} = a_1 \cdot (T_{amb} - T_{HTF}) - a_2 \cdot (T_{amb} - T_{HTF})^2 + U_{sky} \cdot (T_{sky} - T_{HTF}) + U_{wind} \cdot V_{wind} \cdot (T_{amb} - T_{HTF}) + \dot{m} \cdot Cp \cdot \frac{N}{A_{ref}} \cdot (T_{in} - T_{HTF}) + \eta_{opt} \cdot G_b \tag{6}$$

where:

- $C_{col}$ (J$\cdot$m$^{-2}$) is heat capacity of the collector per unit surface area;
- $U_{sky}$ (W$\cdot$m$^{-2} \cdot$ K$^{-1}$) is sky temperature dependence of the heat loss coefficient;
- $T_{sky}$ (C) is sky temperature;
- $U_{wind}$ (J$\cdot$m$^{-3} \cdot$ K$^{-1}$) is the wind speed dependence of the heat loss coefficient;
- $V_{wind}$ (m $\cdot$ s$^{-1}$) is the mean wind speed;
- $\dot{m}$ (kg$\cdot$ s$^{-1}$) is the mass flow rate.

The optical efficiency $\eta_{opt}$ is modified to simulate the performance of LFC as [8]:

$$\eta_{opt} = \eta_0 \cdot IAM(\theta_L) \cdot IAM(\theta_L) \cdot \eta_{endlosses} \cdot \eta_{cleanness} \cdot \eta_{tracking} \tag{7}$$

where:

- $\eta_{endlosses}$ is the optical losses coefficient;
- $\eta_{tracking}$ is the tracking error coefficient.

The *IAM* coefficients are factorised in the *CARNOT* model, which is not the case for *RealTrackEff*. The end losses efficiency coefficient $\eta_{endlosses}$ is defined as:

$$\eta_{endlosses} = 1 - tan(\theta_i) \cdot \frac{X}{L_{receiver}} \tag{8}$$

where:

- $\theta_i$ is the incidence angle;
- $X$ (*m*) is the mean distance between primary mirrors and the receiver;
- $L_{receiver}$ (*m*) is the length of the receiver tube.

### 2.1.4. The Oil Loop Elements

In order to simulate the operation of the HTF loop and not only the LFC via the 3 aforementioned models, a heat-exchanger and a tank have been considered. Thus, the hot fluid leaving the receiver on the north edge goes to a 300 L ($V_0$) tank, where the oil can expand with the temperature (as seen in Figure 4). The system is pressurised with the help of Nitrogen gas. The tank is also used also as a buffer with a wired wrapped-around heater, to maintain the oil inside with a temperature level of at least 70 °C . This is a requirement stemming from the pump in use due to viscosity consideration. The tank operation is as follows:

$$\begin{cases} T_{out} = \frac{V_{in} \cdot T_{in} + (V_{tank,k-1} - V_{out}) \cdot T_{tank,k-1}}{V_{in} + V_{tank,k-1} - V_{out}} \\ V_{out} = \frac{\dot{m}}{\rho_{out}} \end{cases} \tag{9}$$

where:

- $V_{in}$ (m$^3$) is the volume of the HTF entering into the tank;
- $T_{tank,k}$ (m$^3$) is the average temperature of the tank at the step $k$;
- $V_{tank,k}$ (m$^3$) is the volume of the tank at the step $k$, where $V_{tank,0} = V_0 = 0.3$ m$^3$.

The heat-exchanger is regulated by a thermostat function. Heat-exchange is triggered once the temperature of the tank reaches 165 °C and stops when the temperature reaches 120 °C . The maximum decrease in oil temperature allowed by the heat-exchanger is 20 °C between the inlet and outlet, with a maximum decrease of 1 °C per minute.

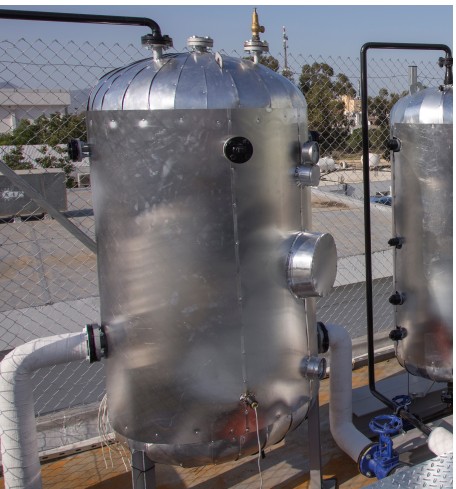

**Figure 4.** Buffer tank of oil with electric heaters wrapped around, allowing the expansion of the oil under pressure with Nitrogen gas.

### 2.1.5. Comparison

The 3 models have been compared to the real outlet temperature. More specifically, the data from the 8th of August 2019 are illustrated in Figure 5. The weather conditions are presented in Figure 6, namely the DNI and the ambient temperature along the day.

As can be seen, in yellow, the ISO-based outlet temperature overestimates the real outlet temperature (in red). The CARNOT outlet is represented in green. As can be observed, the model underestimates the temperature of the outlet especially at the beginning of the day before 11 a.m. The ReakTrackEff methodology fits better the real outlet temperature. The main inaccuracy occurs before 7 a.m. in the morning when the outlet temperature is underestimated, but at this moment of the day, with a low level of DNI, modelling accuracy is less relevant. The RMS (root-mean square) for the ISO9806 model is 6.3 °C for this specific day. The RMS for the RealTrackEff is 1.0 °C and the CARNOT model has an RMS of 2.9 °C.

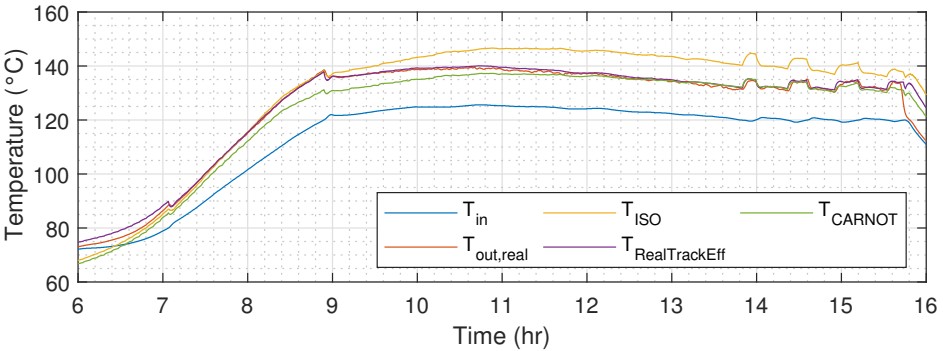

**Figure 5.** Outlet temperatures simulated, compared to the measured temperatures.

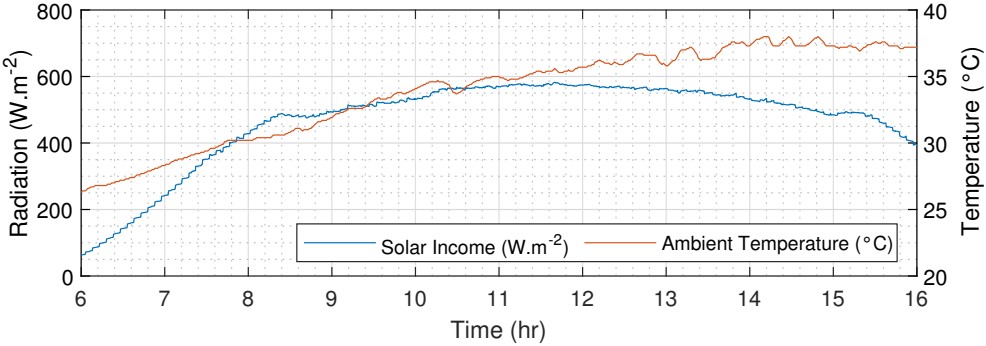

**Figure 6.** Weather conditions.

## 2.2. Controller

The selected controller is a PID. It is as described in Figure 7, including an anti-windup. It has been tuned manually. The working environement is Matlab/Simulink.

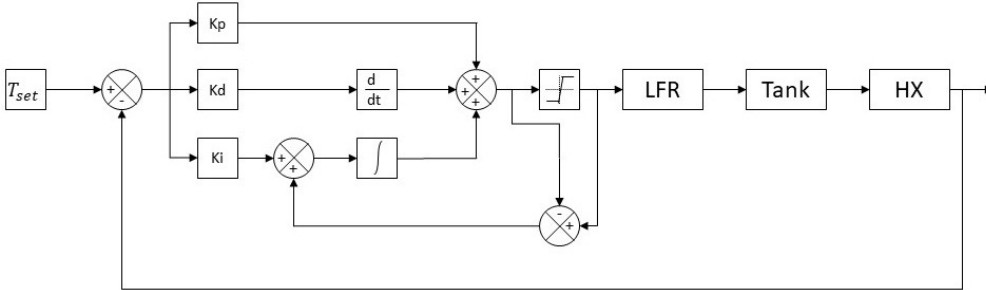

**Figure 7.** Schematics of the PID and the HTF loop components.

The Reynolds number (*Re*) of the flow in the absorber shall be higher than 10,000. This constraint is imposed by the manufacturer of the collector in order to ensure a turbulent state in the absorber. From a practical point of view in operation, if the flowrate is too low, the circurmference of the aborber wall is not evenly heated up by the HTF, and the absorber pipe may bend vertically toward the glass. This may either cause the steel to be in contact

with the borosilicate glass (which would break instantaneously) or stress the connecting flanges (which will open and oil may leak, which is a source of hazard). Thus, the minimal mass flow into the absorber is defined as $\dot{m}_{min}$:

$$\dot{m}_{min} >= \frac{\mu_{HTF} \cdot D \cdot \pi \cdot Re}{4 \cdot \rho_{HTF}}, \; Re = 10,000 \tag{10}$$

where:

- $\mu_{HTF}$ (kg $\cdot$ m$^{-1}$ $\cdot$ s$^{-1}$) is the dynamic viscosity, as specified by the HTF manufacturer;
- $D$ is the inner diameter of the absorber, which is 66 mm;
- $\rho_{HTF}$ (kg $\cdot$ m$^{-3}$) is the density of the HTF.

The dynamic viscosity $\mu$ is given as function of temperature T ($^{\circ}$C) as:

$$\mu(T) = 1.13 \cdot 10^{-2} - 3.12 \cdot 10^{-4} \cdot T + 4.21 \cdot 10^{-6} \cdot T^2 - 2.96 \cdot 10^{-8} \cdot T^3 + 1.03 \cdot 10^{-10} \cdot T^4 - 1.40 \cdot 10^{-13} \cdot T^5 \tag{11}$$

As can be seen in Figure 8, the minimum allowed flowrate to respect the condition *Re* greater than 10,000 decreases dramatically with the increase in temperature. When the heat-exchange is activated, the minimum flowrate is 0.8 L $\cdot$ s$^{-1}$, and it decreases down to 0.4 L $\cdot$ s$^{-1}$ at 165 $^{\circ}$C . The minimum flow rate limits the temperature increase of the oil. However, at low temperature, the minimum flow rate is much higher. The PID includes an anti-windup based on the minimal flow as defined above and a maximum flowrate of 20 L $\cdot$ s$^{-1}$. The PID has been manually set up for the RealTrackEff. The inlet temperature of LFC was set to 120 $^{\circ}$C with a reference for the outlet of 125 $^{\circ}$C . The weather condition taken into account are the ones of the 8 August 2019 at 11 a.m., which are presented in Figure 9. More specifically, the ambient temperature was 34.9 $^{\circ}$C, and the DNI was 935 W $\cdot$ m$^{-2}$. The response time at 99% was 32.4 s. Applying the same PID tuning on the CARNOT model, the response time would be shorter by 9.3%, i.e., 29.4 s. On the other hand, the ISO9806 model presents a slight overshoot of 4% and a much longer time response of 53.8 s (+66%), as observed in Figure 10.

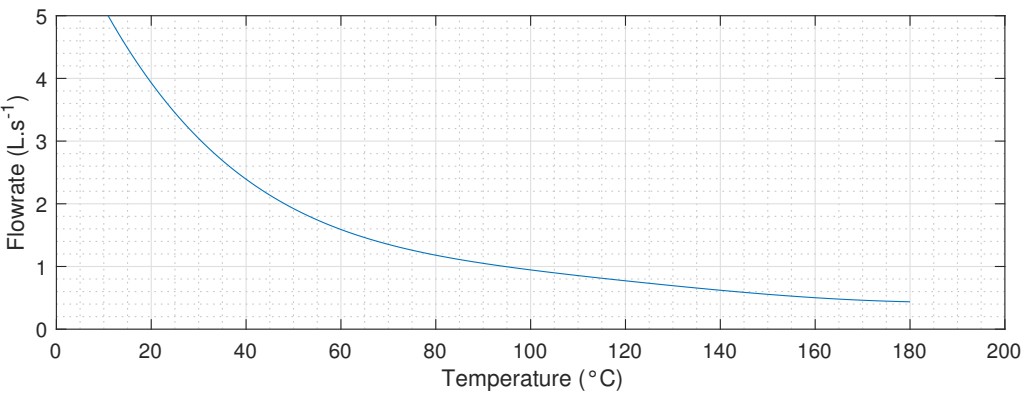

**Figure 8.** Minimum allowed flow of the heat transfer fluid.

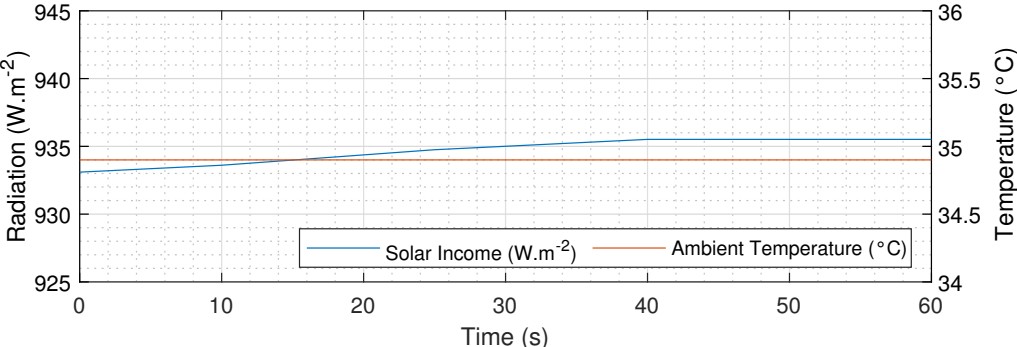

**Figure 9.** Weather data on the 8 August 2019 for the PID tuning.

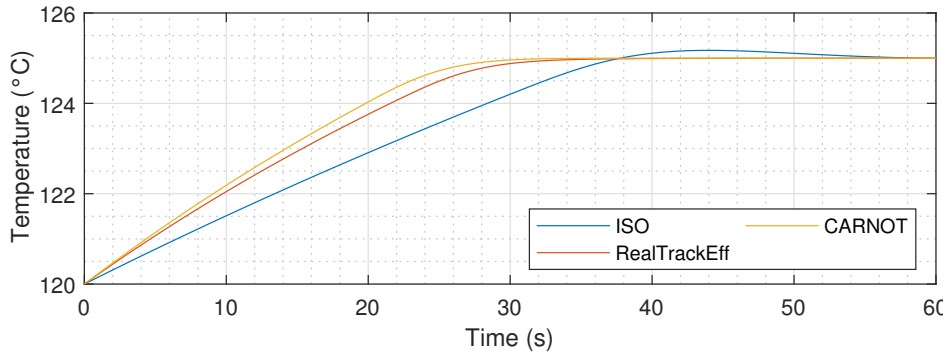

**Figure 10.** Outlet temperature of the HTF of the LFC on the 8 August 2019 for the PID tuning.

The flows are displayed in Figure 11. As can be seen, the flow saturates at the start as the flow rate is limited by the minimum *Re* of 10,000 for the 3 models, corresponding to a flowrate of 1.6 L · s. Once the value of the desired flow is above the lower limit, it increases beyond saturation (20.5 s for the CARNOT, 21.8 s for RealTrackEff and 30.8 s for ISO). RealTrackEff and CARNOT see their flow stabilising much faster than the ISO9806 model, which also presents an overshooting at 50 s. This shows that CARNOT and RealTrackEff behave similarly, although the controller has been defined for RealTrackEff. The final value of the stabilised flowrate for RealTrackEff (3.9 L · s$^{-1}$) is lower than CARNOT (4.5 L · s$^{-1}$). The flowrate for the ISO, on the other hand, peaks at 7.1 L · s$^{-1}$, denoting a huge difference with the 2 other models, which may lead to issues when it goes to the operation of the pump. In the next section, the PID will be evaluated for the entire oil loop including the heat-exchanger and the tank. However, already major differences are noted between the models especially with the ISO9806 only focusing on the receiver itself.

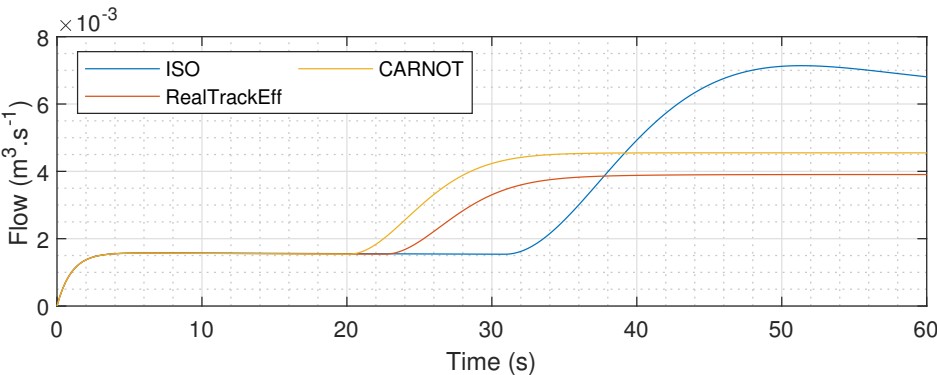

**Figure 11.** Flow of the HTF on the 8 August 2019 for the PID tuning.

## 3. Results

The PID has been applied to real weather data for the same day (8 August 2019) for one hour starting at 11 a.m. The weather data ($G_b$ and $T_{amb}$) are displayed in Figure 12. The DNI oscillated around 935 W · m$^{-2}$ and the temperature stayed around 35 °C. The set temperature of the outlet HTF temperature is 180 °C. The effective simulated outlet temperature of the HTF is displayed in Figure 13, while the flow rate is displayed in Figure 14. The CARNOT model reaches a maximum temperature of 173.7 °C at 0.18 h (10 min). The RealTrackEff modelling reaches a maximum temperature of 180 °C at 0.2 h (12 min), where it stagnates on a plateau for almost 2 min. As previously defined, once the average temperature of the tank reaches 165 °C, the heat-exchanger is activated. As the CARNOT model heats up faster, logically, the heat-exchange happens faster. This can be seen with the average temperature of the tank in Figure 14. The average temperature of the tank reaches the threshold of 165 °C faster with the CARNOT model than with the 2 other models. RealTrackEff reaches 165 °C in its average temperature of the tank 2 min after the CARNOT. As can been observed in Figure 14, the CARNOT flowrate is much slower on the warming phase than the 2 other models. The outlet

temperature does not reach the set temperature of 180 °C. Thus, after the warming phase, the flowrate increases up to its stabilised value of 1.21 L · s$^{-1}$ at 29 min.

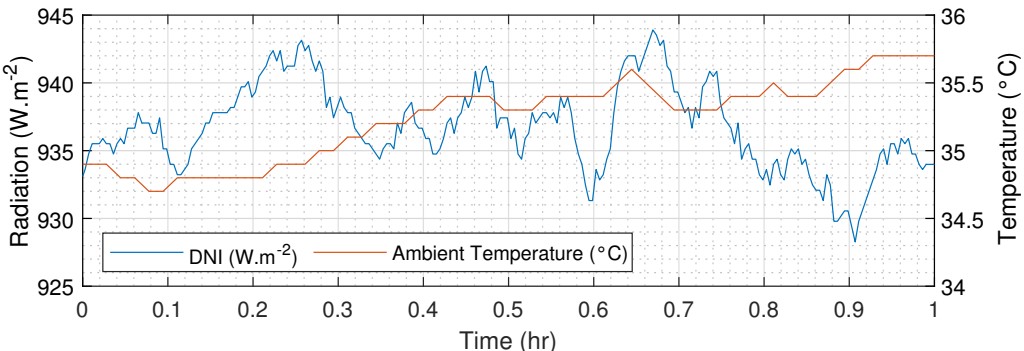

**Figure 12.** Weather conditions on the 8 August 2019 at 11 a.m.

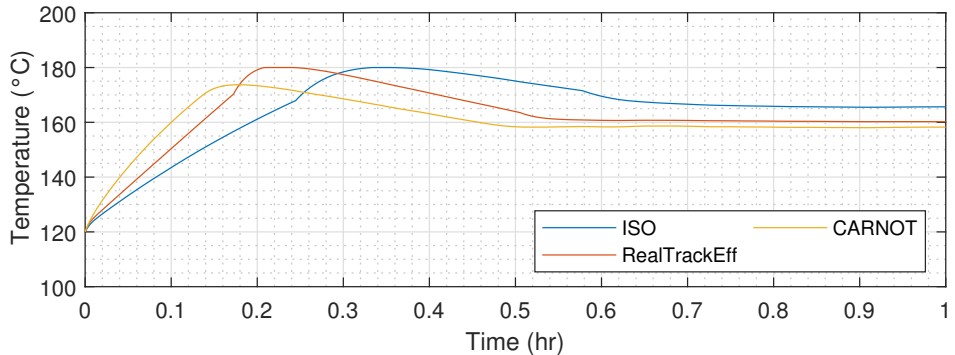

**Figure 13.** Outlet temperature of the HTF with the 3 modellings.

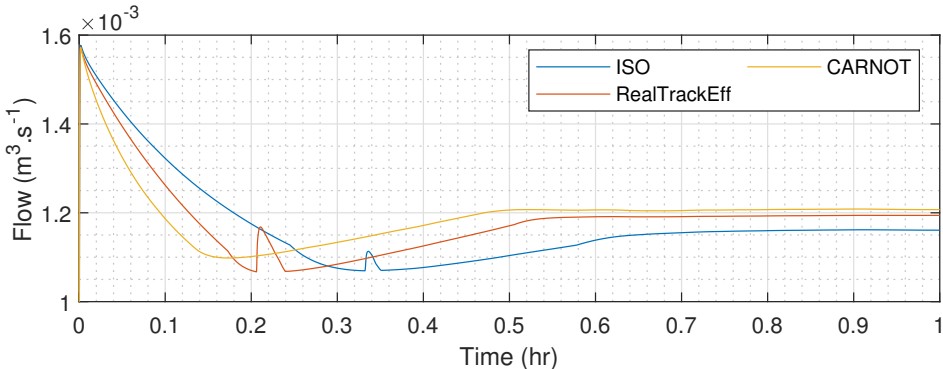

**Figure 14.** Flow of the HTF with the 3 modellings.

On the other hand, for RealTrackEff, the activation of the heat-exchanger is delayed, and the effect of the opening does not decrease the average temperature of the tank as quickly as for CARNOT. Slightly after 12 min, the flowrate increases as the outlet temperature reached the set temperature of 180 °C. Once the temperature of the HTF decreases in the tank after the activation of the heat-exchange, meaning that the power extracted by the heat-exchanger is superior to the heat produced by the LFC, the flow rate decreases. Then, it increases until it reaches a stabilised value of 1.19 L · s$^{-1}$ at 34 min, which is slightly lower than for the CARNOT, which was 1.21 L · s$^{-1}$. Again, CARNOT and RealtrackEff are responding similarly with the PID.

As can be observed in Figure 15, the average tank temperature reaches almost the same stabilised value for the RealTrackEff and CARNOT (respectively 160.2 °C and 158.2 °C), but for the ISO9806 model, the value is higher (165.6 °C). Regarding the ISO9806 modelling, which is the most commonly used, the results are quite different. It is much

slower than the two other ones in reaching the value of 165 °C as a tank average temperature (14 min). It also stagnates for a short period of time as the outlet temperature reaches 180 °C. The flow rate value also increases slightly as for the RealTrackEff after the warming-up phase. The flow rate starts to decrease once the heat extracted by the heat-exchanger is superior to the heat produced by the LFC. However, the flowrate stabilises much later than for the other models at 48 min with a lower value (1.16 L $\cdot$ s$^{-1}$). Regarding the instantaneous power, as displayed in Figure 16, the power is limited to a plateau value for the 3 models in the warming-up phase due to the flow rate limitation. The longer the warming-up phase, the longer the plateau. Once the heat-exchange is activated, the power increases dramatically but in different proportions. The CARNOT modelling's power increases to 26 kW, the RealTrackEff modelling's power increases to 42 kW, and that of the ISO increases to 57 kW in the cooling phase. In the stabilised phase, the respective powers are 24 kW (−31% compared to RealTrackEff), 35 kW and 46 kW (+32% compared to ReaTrackEff).

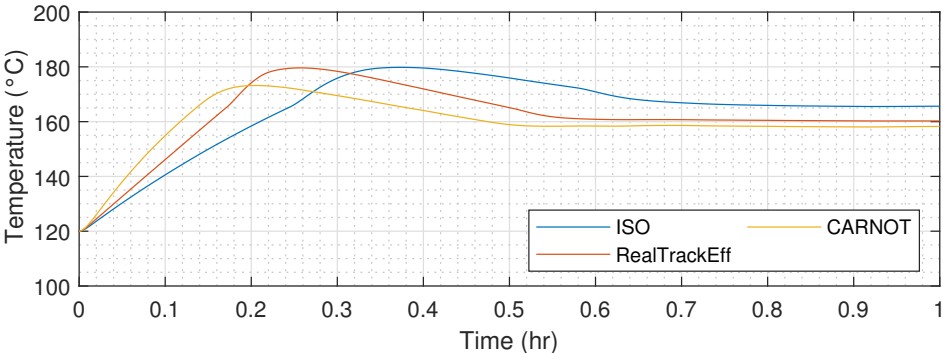

**Figure 15.** Tank average temperature of the 3 modellings.

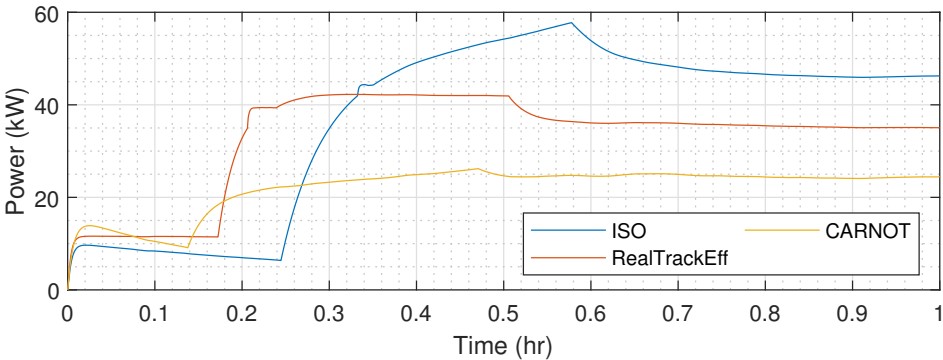

**Figure 16.** LFC power of the 3 modellings.

## 4. Conclusions

The thermal performance test model is a method to define the offline quasi-dynamic physical model according to the energy balance theory and the heat-transfer principle based on empirical methods. The ISO9806 is one of the performance testing models which is often used to predict the quasi-dynamic behaviour of the linear Fresnel reflectors. However, it presents some flaws when applied to real experimental values as it does not consider the asymmetric behaviour of the collector in terms of *IAMs*. Another alternative modelling studied was the CARNOT, which is an open-source toolbox for MATLAB/Simulink. However, it also uses the factorisation of the *IAMs* such as the ISO9806.

This paper presented a comprehensive and efficient approach to model LFCs by improving the ISO9806, using real data extracted from the plant located at the Cyprus Institute in Nicosia, Cyprus. In order to maintain the outlet temperature of the solar collector at a specified value, a PID controller with anti-reset windup has been applied.

The proposed model RealTrackEff has been compared to the ISO9806 and the CARNOT models, both with and without applying the PID controller. The results showed that the RealTrackEff provides the best fitting model with the real outlet temperatures with an accuracy of 1.0 °C (against 2.9 °C for the CARNOT and 6.3 °C for the ISO9806). Thus, it is considered as the most realistic and appropriate model for the tuning of the controller's parameters.

By applying a PID controller, the results in temperature differ a lot, especially with the ISO9806 and somewhat less with the CARNOT. However, as far as the power is concerned, the differences stray dramatically between −31% and 32%. The PID applied to the CARNOT loop leads to much lower power output. Once applied to the ISO9806, the same control overestimates the power output, which is the most relevant value for annual yield predictions.

These observations demonstrate the accuracy and reliability of the proposed model that will be of higher importance when considering annual calculations, which matter in terms of energy planning.

**Author Contributions:** Conceptualization, A.C.M. and R.M.; methodology, A.C.M. and R.M.; software, A.C.M. and R.M.; validation, A.C.M. and R.M.; formal analysis, A.C.M.; investigation, A.C.M.; resources, A.C.M.; data curation, A.C.M.; writing—original draft preparation, A.C.M. and R.M.; writing—review and editing, A.C.M. and R.M.; visualization, A.C.M.; supervision, A.C.M.; project administration, A.C.M.; funding acquisition, A.C.M. All authors have read and agreed to the published version of the manuscript.

**Funding:** The research that led to these results has received funding from the European Union's Horizon 2020 research and innovation programme under grant agreements No 823802 (SFERA-III).

**Institutional Review Board Statement:** Not applicable.

**Informed Consent Statement:** Not applicable.

**Acknowledgments:** The authors would also like to acknowledge the technical support of Nicolas Jarraud.

**Conflicts of Interest:** The authors declare no conflict of interest.

## Abbreviations

The following abbreviations are used in this manuscript:

| | |
|---|---|
| DNI | Direct Normal Irradiance |
| HTF | Heat Transfer Fluid |
| IAM | Incidence Angle Modifier |
| LFC | Linear Fresnel Collector |
| PID | Proportional–Integral–Derivative |
| RMS | Root Mean Square |

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
