# Peer review of "Control Strategies Applied to a Heat Transfer Loop of a Linear Fresnel Collector"

_energies, doi:10.3390/en15093338_

Round 1

Reviewer 1 Report

Review of “Control strategies applied to a heat transfer loop of a Linear Fresnel Collector”

Alaric Christian Montenon and Rowida Meligy

This is an interesting paper, particularly to other researchers working on simulation.

The problem is well set out and the results well-presented and discussed.

There are some areas where I feel the paper would benefit from more detail and discussion.

  • Figure 7 and the surrounding text describe the flow rate require to achieve Re=10,000 and how this changes significantly with temperature. Since changing the flow rate will significantly affect the thermal performance, it is not clear why this restriction in Re is enforced. It would also be beneficial to make it explicitly clear the extend of temperature variation which is present in standard operating conditions.
  • There is significant variation between the models in many of the figures and this leads the authors to conclude “The work done here demonstrated the sensitivity of the modelling to the control that will be then amplified when considering annual calculations.” What would be useful to the reader would be some insight into what are the key differences between the models, how the operate and how they perform. Also, some indication how and when the different models can be best applied.

The paper is generally well written; however, there is sometimes a non-standard choice of words e.g. “Re superior to 10,000” instead of “Re greater than 10,000”. The manuscript would benefit from a careful proof read.

Author Response

Dear reviewer, please see our response in the attached pdf.

Reviewer 2 Report

Dear Authors,

In the Abstract section, please state the "gaps" mentioned in the ISO-9806 standard. Also, describe the achievements and optimizations made by your research work more clearly and quantitatively to justify the necessity of doing this research.

Some citations in the introduction and specially lines 1-9 seem unnecessary (or poorly done like line 38). For a proper reference in a scientific article, it should, in a way, mention what the purpose of the destination work is and also try to refer to the State-of-the-art works, especially in the introduction section.

Figures 1, 2, 6, 8, 9, 11, 14 and 15 are located before referencing it in the context, not recommended.

I also suggest that the introduction section (and some other parts outside introduction) be reviewed and processed in a coherent manner and clearly define and discuss the stages of your work. There are short sentences in a row that make it difficult to follow for reader. Also lines 60-62 are not suitable and need rewriting.

2.1. Modelling: I suggest you consider a table for inserting tube information that includes material and diameter and thickness information. Of course, along with the construction standard. Also, I suggest presenting the process flow diagram of the research work.

Line 178 = There is not any flow rate data in Figure 12, probably typo for Figure 13, please fix it. The same comment for line 183. 

Some mistakes in sentences need to be corrected, such as line 200 ("but the for the"). The same comment for other parts of the paper.

The article does not have a conclusion section, please add this section and provide the results.

Kind regards,

Author Response

(The authors gave the same response as above.)

Reviewer 3 Report

The authors presented two alternative modellings methodologies of a Linear Fresnel Collector based on recorded experimental data.

The sketch of the study lacks any new idea behind, and is very simple and old in terms of idea. Hence, I do not recommend the study for possible publication.

The manuscript’s strengths.

1) The objective of the study matches the scope of the journal.

The manuscript’s weaknesses.

1) The paper needs better justification of sufficient novelty. Please specify your major research contribution and benefits. In my opinion there is no novelty in the paper.

2) The aim of the study is nuclear.

3) The manuscript is not well organised and not easy to read.

4) Used self-citations are excessive.

Author Response

(The authors gave the same response as above.)

Reviewer 4 Report

Thanks for inviting me to review this paper “Control strategies applied to a heat transfer loop of a Linear Fresnel Collector”. In this paper, two different models are constructed. Both models include all the elements of the heat transfer fluid loop. All the results are compared and a PID control is applied on the experimental values in order to estimate the impact on the outlet temperature on the absorber and on power generation. My suggestions or questions are as follows:

  1. There are many parameters in this paper, you’d better write a list of parameters in front of the paper.
  2. Some parameters in this paper have no units, please check it.
  3. There should be more subheadings in result part, which can make the structure clearer and the content richer.
  4. The discussion part is too short, and it’s also not deep enough.
  5. Please check your English language.

Author Response

Dear reviewer,

Thanks for your constructive comments. I have received this review at the same time as your second one.

Thus, we have acknowledged the last one.

Best regards

Round 2

Reviewer 2 Report

Dear Authord,

Thank you for your answers and the changes you made to the text. My remaining advice is to make the article as easy as possible for the reader to read, for example on pages 3-4 and 6-7, images and tables appear in the middle of the text (and paragraphs) which is not recommended.

I suggest you rewrite the conclusion section by restating the necessity of doing the research, summarizing the main points and state the significance of results and then conclude.

Kind regards,

Author Response

Dear reviewer,

Thanks again for your time in reviewing the paper and your helpful comments.

Regarding the locations of the images, thanks for pointing this error out. We have modified them in page 7, but we couldn't identify the issue on pages 3,4 and 6.

The conclusion section has been rewritten and it appears in blue in the new manuscript.

Best regards

Reviewer 3 Report

The sketch of the study lacks any new idea behind, and is very simple and old in terms of idea. Hence, I do not recommend the study for possible publication. 

Author Response

Dear reviewer,

We thanks for your time in reviewing and again we still believe that the RealTrackEff is an advance for the the modelling of the quasi-dynamic operation of LFRs. For annual yield modelling, it is of higher importance with the application of a proper controller as demonstrated in the present paper.

Best regards

Reviewer 4 Report

OK. I think it is acceptable now.

Author Response

Dear reviewer,

Thanks for your time in reviewing our paper and contributing in enriching it.

Best regards

Round 3

Reviewer 3 Report

The authors presented two alternative modellings methodologies of a Linear Fresnel Collector based on recorded experimental data. The sketch of the study lacks any new idea behind, and is very simple and old in terms of idea. The paper has not been improved much compared with the original version. Hence, I do not recommend the study for possible publication.